# HiFA: High-fidelity Text-to-3D generation with Advanced Diffusion Guidance

**Junzhe Zhu**[*1]**, Peiye Zhuang**[*1,2]**, Sanmi Koyejo**[1]
[1]Stanford University,  [2]Snap Inc.
[1]{josefzhu, peiye, sanmi}@stanford.edu, [2]pzhuang@snapchat.com

## ABSTRACT

The advancements in automatic text-to-3D generation have been remarkable. Most existing methods use pre-trained text-to-image diffusion models to optimize 3D representations like Neural Radiance Fields (NeRFs) via latent-space denoising score matching. Yet, these methods often result in artifacts and inconsistencies across different views due to their suboptimal optimization approaches and limited understanding of 3D geometry. Moreover, the inherent constraints of NeRFs in rendering crisp geometry and stable textures usually lead to a two-stage optimization to attain high-resolution details. This work proposes holistic sampling and smoothing approaches to achieve high-quality text-to-3D generation, all in a single-stage optimization. We compute denoising scores in the text-to-image diffusion model's latent and image spaces. Instead of randomly sampling timesteps (also referred to as noise levels in denoising score matching), we introduce a novel timestep annealing approach that progressively reduces the sampled timestep throughout optimization. To generate high-quality renderings in a single-stage optimization, we propose regularization for the variance of z-coordinates along NeRF rays. To address texture flickering issues in NeRFs, we introduce a kernel smoothing technique that refines importance sampling weights coarse-to-fine, ensuring accurate and thorough sampling in high-density regions. Extensive experiments demonstrate the superiority of our method over previous approaches, enabling the generation of highly detailed and view-consistent 3D assets through a single-stage training process.

## 1 INTRODUCTION

The task of automatic text-to-3D generation aims to create 3D assets based on a text description and has gained significant attention due to its wide-ranging applications in digital content generation, film-making, and Virtual Reality (VR) (Lin et al., 2023; Chen et al., 2023b). Initial efforts in this domain centered on unconditional 3D asset generation, experimenting with various 3D representation modalities presented in explicit formats such as meshes (Achlioptas et al., 2018; Luo & Hu, 2021; Smith & Meger, 2017; Xie et al., 2018), as well as implicit formats such as fields (Chen & Zhang, 2019; Mittal et al., 2022; Zhuang et al., 2023). Following this, the field has progressed towards conditional 3D generative models, e.g., with text-based guidance (Cheng et al., 2023). However, these studies have been limited to relatively simple 3D assets, primarily due to the scarcity of large-scale annotated 3D datasets.

The availability of ample image datasets and the success of text-to-image generation have paved the way for lifting pre-trained text-to-image models to the 3D domain. Specifically, recent studies focus on optimizing a 3D representation for an asset, using pre-trained text-to-image generative models by providing a denoising score for rendered images (Khalid et al., 2022; Jain et al., 2022; Poole et al., 2022; Xu et al., 2022; Wang et al., 2023a; Lin et al., 2023; Tang et al., 2023; Chen et al., 2023b; Wang et al., 2023b). Poole et al. (2022) proposed a loss from the distillation of a text-to-image diffusion model. They minimized the Kullback-Leibler (KL) divergence between a family of Gaussian distributions based on the forward diffusion process and the denoising scores acquired from the pre-trained text-to-image diffusion model. The proposed Score Distillation Sampling (SDS) method combined with a NeRF enables 3D asset generation from given text prompts. Subsequent

---

*Equal contribution

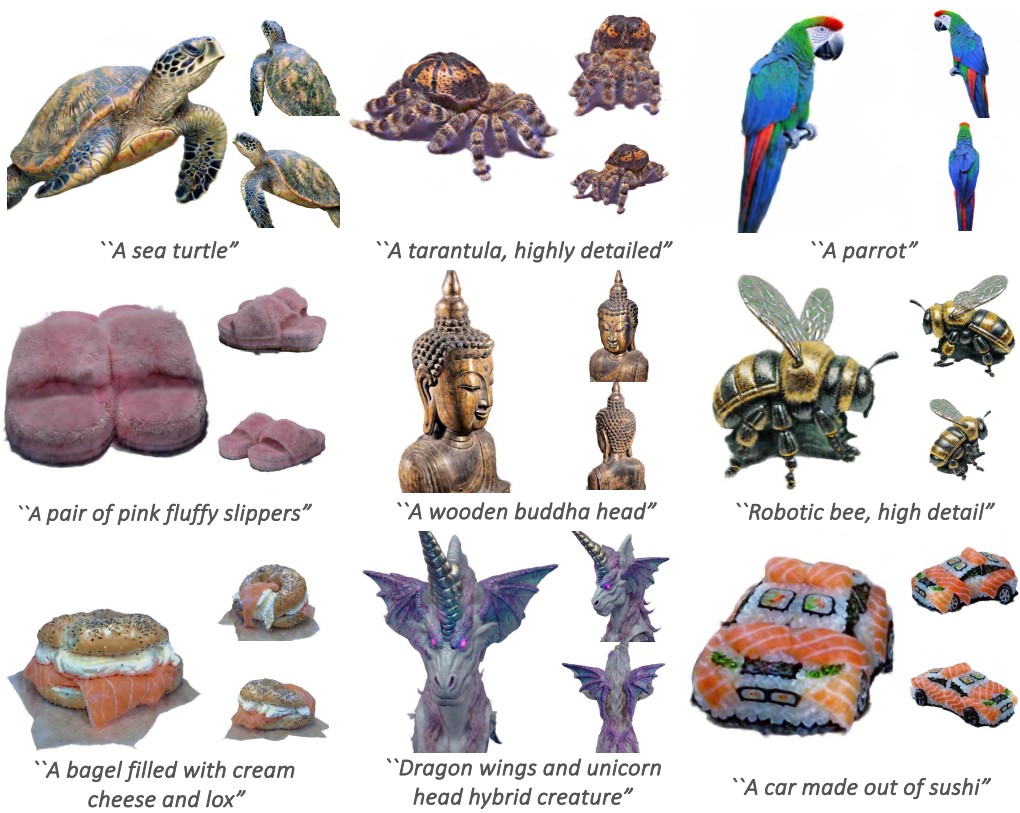

``A sea turtle''    ``A tarantula, highly detailed''    ``A parrot''

``A pair of pink fluffy slippers''    ``A wooden buddha head''    ``Robotic bee, high detail''

``A bagel filled with cream cheese and lox''    ``Dragon wings and unicorn head hybrid creature''    ``A car made out of sushi''

Figure 1: **Examples of multiple views of 3D objects generated by from our model given text prompts (below each object).**

research has improved generation quality through various approaches including the adoption of two-stage optimization frameworks (Lin et al., 2023; Tang et al., 2023; Wang et al., 2023b; Chen et al., 2023c), alterations to the original SDS formulation (Wang et al., 2023a;b), and the disentanglement of geometry and appearance (Chen et al., 2023b).

In this work, we revisit the integration of the SDS approach with NeRFs, aiming to achieve photo-realistic and high-quality text-to-3D generation through a *single-stage* optimization. In contrast to existing text-to-3D generation work, we distill the score in the text-to-image diffusion model's latent and image spaces for enhanced supervision. Moreover, we observe that the efficacy of the diffusion prior is limited in previous works (Poole et al., 2022; Lin et al., 2023) when timesteps (also referred to as noise levels in denoising score matching) are randomly sampled during optimization. Specifically, we observe that toward the end of the training process, the NeRF becomes *almost determined* in representing a particular 3D asset. Thus, we find that randomly sampling a large timestep drives the diffusion model to produce a denoised image that is *distinct* and *unrelated* to the original input rendering. This yields inconsistent distillation from the diffusion model and compromised optimization of NeRFs. To address this, we introduce a timestep annealing approach where the timestep in the forward diffusion process inversely correlates with the square root of the number of training iterations. Our empirical analysis demonstrates that the proposed timestep annealing approach effectively enhances generation quality. We also show that the square root timestep annealing consistently outperforms other annealing methods, such as linear and cosine ratios.

Moreover, generating a detailed 3D asset through single-stage optimization is challenging. Specifically, explicit 3D representations, such as meshes, struggle to capture intricate topology, such as those with holes. Implicit 3D representations (Mildenhall et al., 2020; Müller et al., 2022) may lead to cloudy geometry and flickering textures. For instance, when NeRFs are employed to represent highly detailed 3D geometries like human bodies (Hong et al., 2023), Moiré patterns are noticable. To this end, prior works (Lin et al., 2023; Wang et al., 2023b) adopted two-stage optimization techniques. In these approaches, explicit 3D representations, such as Deep Marching Tetrahedra (DMTet) (Shen et al., 2021), are used to extract textured meshes from the implicit representations in

the first stage and are subsequently fine-tuned in the second stage to capture high-quality geometry. However, these mesh representations forfeit the ability to produce detailed appearances such as fur – a tradeoff we wish to avoid. Differently, we aim to maintain the flexibility and photo-realism offered by the NeRF representation while at the same time achieving high-quality text-to-3D generation through a *single-stage* training.

To this end, we propose two techniques to advance NeRF optimization. Specifically, to address the cloudy geometry issue in NeRFs, we propose a variance regularization that minimizes the variance of sampled z-coordinates distributed along NeRF rays. We observe that this technique enables NeRFs to more accurately represent crisp geometrical surfaces, thereby effectively mitigating the cloudiness issue. Additionally, we verify that our proposed z-variance regularization outperforms alternative spatial regularization proposed in previous methods (Barron et al., 2022).

Moreover, texture flickering or shimmer effects often result from inaccuracies in estimating the importance sampling weights across different rendering views. However, existing solutions, such as increasing the number of sampling points along the rays or deploying separate density estimation networks for each coarse and refined stage, come with increased computational demands. Instead, we propose a kernel smoothing technique tailored for coarse-to-fine importance sampling along NeRF rays without an increase in the computational cost. This technique is inspired by the integrated positional encoding for spatial points within a cone, previously proposed to tackle aliasing issues in a single image view (Barron et al., 2021). In our case, the goal is to mitigate flickering issues across multiple views. Specifically, we use a kernel to refine the probability density function (PDF) estimated in the coarse sampling stage along a ray, which enables more comprehensive sampling near asset surface regions in the refined stage. This technique notably enhances the fidelity of importance sampling.

We summarize our technical contributions for two crucial components of text-to-3D generation: (1) 3D representation and (2) optimization, which are outlined below:

- To achieve photo-realistic and highly-detailed text-to-3D generation, we propose score distillation in both the latent and image space of the pre-trained text-to-image diffusion models. Moreover, we introduce a timestep annealing approach for score distillation from text-to-image diffusion models.

- To achieve sharp geometry quality through a *single-stage* training, we present a regularization method applied to the variance of z-coordinates along NeRF rays.

- To address flickering issues in NeRFs, we propose a kernel smoothing technique that refines the PDF estimation in coarse-to-fine importance sampling.

Taken together, we show how these holistic modifications address existing shortcomings and improve the quality of 3D synthesis.

## 2 RELATED WORK

**Unconditional 3D asset generation** involves the learning of 3D asset data distributions. Explicit approaches employ representations including point clouds (Achlioptas et al., 2018; Luo & Hu, 2021), voxel grids (Lin et al., 2022; Smith & Meger, 2017; Xie et al., 2018) and meshes (Zhang et al., 2021). In contrast, implicit methods utilize representations such as signed distance functions (SDFs) (Chen & Zhang, 2019; Cheng et al., 2023; Mittal et al., 2022), tri-planes (Chen et al., 2023a), multi-layer perceptron (MLP) weights (Erkoç et al., 2023), and radiance fields (Lorraine et al., 2023). However, due to the limited availability of diverse 3D assets, these works primarily focus on generating class-specific and small-scale 3D datasets.

**Text-to-3D asset generation** refers to the creation of 3D assets based on text descriptions. Instead of depending on limited text-annotated 3D datasets, the availability of ample text-image data pairs and the success of text-to-image generative models have inspired recent research to lift pre-trained text-to-image models into the 3D domain. Generally, these approaches can be categorized into two groups: (i) CLIP-guided text-to-3D approaches (Khalid et al., 2022; Jain et al., 2022) that utilize pre-trained cross-modal matching models like CLIP (Radford et al., 2021), and (ii) 2D diffusion-guided text-to-3D approaches (Poole et al., 2022; Tang et al., 2023; Lin et al., 2023; Chen et al., 2023b) that rely on text-to-image diffusion-based generative models such as Imagen (Saharia et al., 2022) and StableDiffusion (Rombach et al., 2022). We follow the diffusion-guided methods due to their superior performance in text-to-3D generation.

Specifically, Poole et al. (2022) first introduced a Score Distillation Sampling (SDS) approach, where noise is added to an image rendered from NeRFs and subsequently denoised by a pre-trained text-to-image generative model (Saharia et al., 2022). SDS minimizes the KL divergence between a prior Gaussian noise distribution and the estimated noise distribution. SDS is widely adopted in follow-up works (Lin et al., 2023; Tang et al., 2023; Chen et al., 2023b). For example, Score-Jacobian-Chaining (Wang et al., 2023a) proposed a Perturb-and-Average Scoring method to aggregate 2D image gradients of StableDiffusion (Rombach et al., 2022) over multiple viewpoints into a 3D asset gradient. Wang et al. (2023b) introduced a variational formulation of the SDS approach for diverse generation of 3D assets, yet it needs to train a low-rank adaptation (LoRA) (Hu et al., 2022) for each individual 3D asset to provide the score function of the distribution. Moreover, two-stage optimization frameworks are proposed (Wang et al., 2023b; Lin et al., 2023) that initially extract 3D meshes from implicit representations and then fine-tune them in the second stage to achieve high-resolution details. In this work, we propose refining of the SDS approach and improving the implicit representation to achieve high-quality 3D asset generation in a single-stage optimization process.

**Image-to-3D reconstruction** refers to 3D reconstruction from a provided single image. Typically, as proposed in prior works (Zhou & Tulsiani, 2023; Gu et al., 2023; Liu et al., 2023b;a), pre-trained text-to-image diffusion-based models are used to provide a 2D prior via the SDS approach plus an image reconstruction loss. In a two-stage optimization process, Qian et al. (2023) employs 2D and 3D diffusion priors. In the first stage, they optimize a NeRF representation, and in the second stage, they extract a DMTet mesh from the NeRF for fine-tuning. Haque et al. (2023) propose an iterative dataset update strategy for editing NeRFs, leveraging text-to-image diffusion models. Alternatively, Liu et al. (2023c) hallucinates 16 normal view images and directly optimizes a NeRF representation based on them. We extend our work on the image-to-3D reconstruction task and compare to these methods in Sec. 5.4.

## 3 PRELIMINARIES: SCORE DISTILLATION SAMPLING (SDS)

**The SDS approach in diffusion models** is proposed in recent work (Poole et al., 2022) using a pre-trained text-to-image diffusion model to guide the 3D representation parameterized by $\theta$. An image $\boldsymbol{x}$ is generated based on a given camera pose via a differentiable rendering function $g$, denoted as $\boldsymbol{x} = g(\theta)$. The pre-trained text-to-image diffusion model is employed to ensure the rendered images align with its learned image distribution. This work uses a *latent* diffusion model to reduce computational complexity.

Specifically, a *latent* diffusion model such as Stable Diffusion (SD) (Rombach et al., 2022), consists of an encoder $\mathcal{E}$, a decoder $\mathcal{D}$, and a denoising function $\epsilon_\phi$, parameterized by $\phi$. The encoder $\mathcal{E}$ compresses the input image $\boldsymbol{x}$ into a low-resolution latent vector $\boldsymbol{z}$, written as $\boldsymbol{z} = \mathcal{E}(\boldsymbol{x})$. Conversely, the decoder $\mathcal{D}$ reconstructs the image from the latent vector as $\boldsymbol{x} = \mathcal{D}(\boldsymbol{z})$. The denoising score function $\epsilon_\phi$ estimates the given noise as $\hat{\boldsymbol{\epsilon}} := \epsilon_\phi(\boldsymbol{z}_t; \boldsymbol{y}, t)$, where $\boldsymbol{z}_t$ is a noisy latent vector, formally written as $\boldsymbol{z}_t = \alpha_t \boldsymbol{z} + \sigma_t \boldsymbol{\epsilon}$. Here, $\alpha_t$ and $\sigma_t$ define a schedule for adding Gaussian noise $\boldsymbol{\epsilon} \sim \mathcal{N}(\boldsymbol{0}, \boldsymbol{I})$ to the latent vector $\boldsymbol{z}$ given a text embedding $\boldsymbol{y}$ at timestep $t$. Subsequently, the SDS loss is used to provide gradients for optimizing the 3D representation $\theta$, written as

$$\nabla_\theta \mathcal{L}_{\text{SDS}}(\phi, \boldsymbol{z}) = \quad \mathbb{E}_{t, \boldsymbol{\epsilon}}[\omega(t)(\hat{\boldsymbol{\epsilon}} - \boldsymbol{\epsilon}) \frac{\partial \boldsymbol{z}}{\partial \theta}], \tag{1}$$

where $\omega(t)$ is a weighting function.

## 4 APPROACH

We aim to generate high-quality 3D assets in a single-stage approach driven by text prompts. For this, we propose our method as illustrated in Fig. 2. We present our technical contributions in two parts. In Sec. 4.1, we compute the SDS loss in both the latent and image spaces of the pre-trained SD model (Rombach et al., 2022). Unlike previous works (Poole et al., 2022; Lin et al., 2023; Chen et al., 2023b), we propose a simple yet effective timestep annealing approach that gradually reduces timesteps (noise levels) throughout the optimization process. In Sec. 4.2, we introduce our variance regularization loss for z-coordinates along NeRF rays. Additionally, we present a kernel smoothing technique for importance sampling, enabling NeRFs to produce crisp geometry and maintain a view-consistent appearance.

### 4.1 ADVANCING SDS-BASED OPTIMIZATION

**Augmenting the SDS approach in both the image and latent spaces.** In our work, we employ a pre-trained SD model (Rombach et al., 2022) to optimize NeRFs. We extend the score distillation

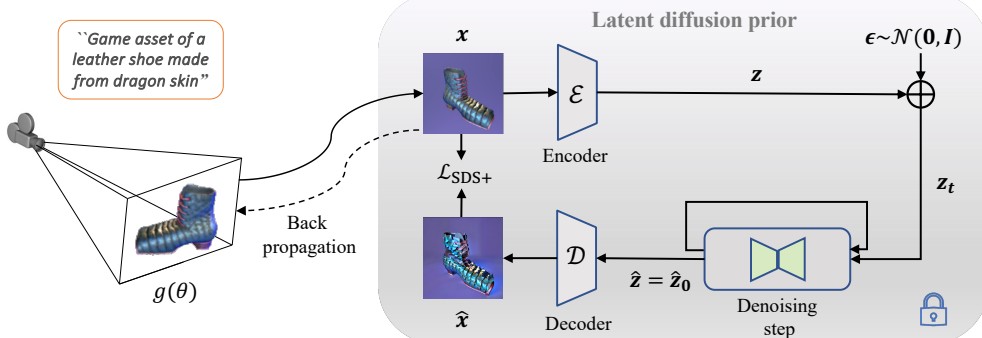

Figure 2: **Overview of our proposed method for text-to-3D generation**. We aim to optimize a 3D model $g(\theta)$ using a pre-trained 2D latent diffusion prior . To achieve this, we employ a latent diffusion model for score distillation. Specifically, the diffusion model takes a rendered image $x$ as input and provides the estimate of the input rendered image, denoted as $\hat{x}$. We utilize $\mathcal{L}_{\text{SDS+}}$ loss that computes reconstruction loss in both the latent and image spaces.

to both the latent and image spaces of the SD model. For this, we first reformulate the original SDS loss (as described in Eq. 1) in terms of the latent vector residual instead of the noise residual:

$$
\begin{aligned}
\nabla_\theta \mathcal{L}_{\text{SDS}}(\phi, z) &= \mathbb{E}_{t,\epsilon}\left[\omega(t)(\hat{\epsilon} - \epsilon)\frac{\partial z}{\partial \theta}\right] \\
&= \mathbb{E}_{t,\epsilon}\left[\omega(t)\left(\frac{1}{\sigma_t}(z_t - \alpha_t\hat{z}) - \frac{1}{\sigma_t}(z_t - \alpha_t z)\right)\frac{\partial z}{\partial \theta}\right] \quad = \mathbb{E}_{t,\epsilon}\left[\omega(t)\frac{\alpha_t}{\sigma_t}(z - \hat{z})\frac{\partial z}{\partial \theta}\right],
\end{aligned}
\tag{2}
$$

where $\hat{z} := \frac{1}{\alpha_t}(z_t - \sigma_t\hat{\epsilon})$ represents the estimate of the latent vector $z$ using the denoising function $\epsilon_\phi$, and $(\hat{z} - z)$ is referred to as the latent vector residual. Note that due to the difficulty of recovering an explicit loss formulation $\mathcal{L}_{\text{SDS}}$ for the gradients in Eq. 1, Poole et al. (2022), directly compute the gradients to optimize 3D representations. In contrast, our reparameterization of the gradients as shown in Eq. 2 allows us to explicitly formulate the $\mathcal{L}_{\text{SDS}}$ loss, thus simplifying the loss visualization and analysis process. Formally, we have

$$
\mathcal{L}_{\text{SDS}}(\phi, z) = \quad \mathbb{E}_{t,\epsilon}\, \omega(t)\|z - \hat{z}\|^2,
\tag{3}
$$

where we incorporate those coefficients related to $t$ into $\omega(t)$.

Subsequently, we further adapt the loss by incorporating supervision for high-resolution images. Formally, we define the adapted loss $\mathcal{L}_{\text{SDS+}}$ as

$$
\mathcal{L}_{\text{SDS+}}(\phi, z, x) = \quad \mathbb{E}_{t,\epsilon}\, \omega(t)\left[\|z - \hat{z}\|^2 + \lambda_{\text{rgb}}\|x - \hat{x}\|^2\right],
\tag{4}
$$

where $\hat{x}$ is an recovered image obtained through the decoder $\mathcal{D}$, formally denoted as $\hat{x} = \mathcal{D}(\hat{z})$ and $\lambda_{\text{rgb}}$ is a scaling parameter. We note that a similar image reconstruction loss is employed in recent image-to-3D reconstruction works (Zhou & Tulsiani, 2023). Our approach is different in two ways. First, we observe that it is inadequate only to use the image residual, i.e., $\|x - \hat{x}\|^2$, without the incorporation of the latent residual $\|z - \hat{z}\|^2$, resulting in color bias issues in text-to-3D generation. We will present the ablation experiments in Sec. 5.2. Second, using a random timestep sampling approach in previous works (Zhou & Tulsiani, 2023; Poole et al., 2022; Lin et al., 2023) during the denoising process limits the guidance of text-to-image diffusion models. In the following, we analyze this in detail and introduce a novel timestep annealing approach designed to enhance the SDS performance.

**A timestep annealing approach** offers a more effective alternative to random timestep sampling used in previous works (Poole et al., 2022; Lin et al., 2023). To be concrete, our observations suggest that random sampling can introduce divergence issues in the denoised images. As training nears completion, a NeRF renders images representing an *almost determined* 3D asset. In this case,

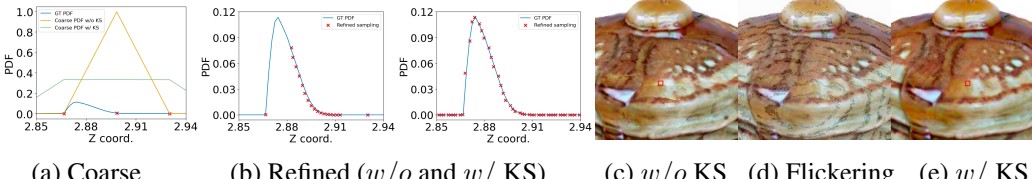

(a) Coarse     (b) Refined ($w/o$ and $w/$ KS)     (c) $w/o$ KS   (d) Flickering   (e) $w/$ KS

Figure 3: **Visualization of the flickering issue.** We display the sampled z-coordinates along the ray for the rendered pixel (marked in the *red square* in (c) and (e)). Specifically, In (a), the ground-truth PDF is shown in blue, while the estimated PDF is shown without (*yellow*) and with (*green*) the kernel smoothing (KS) approach. In (b) we show the sampled z-coordinates in the refined stage, without KS (*left*) or with KS (*right*). Their corresponding rendered image is presented in (c) and (e), respectively. In (d), we overlay the difference of the two renderings (i.e. the flickering) on (c).

if a large timestep $t$ is randomly sampled, the denoising function might predict an image that is *distinct* and *unrelated* to the given input rendering. This can produce inaccurate gradients from the diffusion model, thereby negatively impacting the optimization of the 3D model.

To circumvent this issue, we propose a timestep annealing approach. Specifically, we use a high value of timestep $t$ for the rendered image during the *initial* training iterations. This intentional noise allows the image to align more closely with the distribution characterized by the text-to-image diffusion prior. As training proceeds, we gradually reduce the timestep $t$, thereby capturing finer details through more stable and lower variance gradients.

A question follows: *what is the suitable annealing rate?* We investigated several options, including linear, cosine, and square root schedules. Empirical evaluation (details in Sec. 5) suggests that square root scheduling yields superior results in our scenario, formally written as

$$t = t_{\max} - (t_{\max} - t_{\min})\sqrt{\frac{\text{iter}}{\text{total\_iter}}}, \tag{5}$$

where timestep $t$ decreases steeply during the initial training process and decelerates as the training progresses. This scheduling allocates more training iterations to lower values of timestep $t$, ensuring that fine-grained details are sufficiently captured in the latter iterations of training.

## 4.2 Advancing regularization in NeRF representation

We introduce two techniques to improve NeRF representations, including a regularization method for *the variance of z-coordinates* (a.k.a. z-variance) sampled along NeRF rays and a novel kernel smoothing approach for importance sampling during rendering.

A NeRF renders a pixel color $\hat{C}_r$ of an image, denoted as $\hat{C}_r = \sum_{i=1}^{N} \nu_i c_i$, where $\nu_i$ and $c_i$ are respectively estimated weights and colors of the sampled coordinate $z_i$ along a ray $r$ (Mildenhall et al., 2020). $N$ refers to the number of sampled points along the ray $r$. Accordingly, the depth value $\mu_{z_r}$ and the disparity value $d_{z_r}$ of the ray $r$ are written as

$$\mu_{z_r} = \sum_i z_i \frac{\nu_i}{\sum_i \nu_i}, \text{ and } d_{z_r} = \frac{1}{\mu_{z_r}}, \tag{6}$$

where $\frac{\nu_i}{\sum_i \nu_i}$ can be considered as a sampled PDF.

**Regularization for z-variance** aims to minimize variance in the distribution of sampled z-coordinates $z_i$ along the ray $r$. A reduced variance indicates crisper surfaces in geometry. For instance, in an extreme case where the rendering weights of a ray follow a Dirac delta function, the z-variance will be zero, resulting in an extremely sharp surface. Formally, we denote the z-variance along the ray $r$ as $\sigma_{z_r}^2$:

$$\sigma_{z_r}^2 = \mathbb{E}_z\left[(z_i - \mu_{z_r})^2\right] = \sum_i (z_i - \mu_{z_r})^2 \frac{\nu_i}{\sum_i \nu_i}. \tag{7}$$

The regularization loss $\mathcal{L}_{\text{zvar}}$ for the variance $\sigma_{z_r}^2$ is defined as

$$\mathcal{L}_{\text{zvar}} = \mathbb{E}_r[\delta_r \sigma_{z_r}^2], \quad \delta_r = 1 \text{ if } \sum_i \nu_i > 0.5, \text{ else } 0. \tag{8}$$

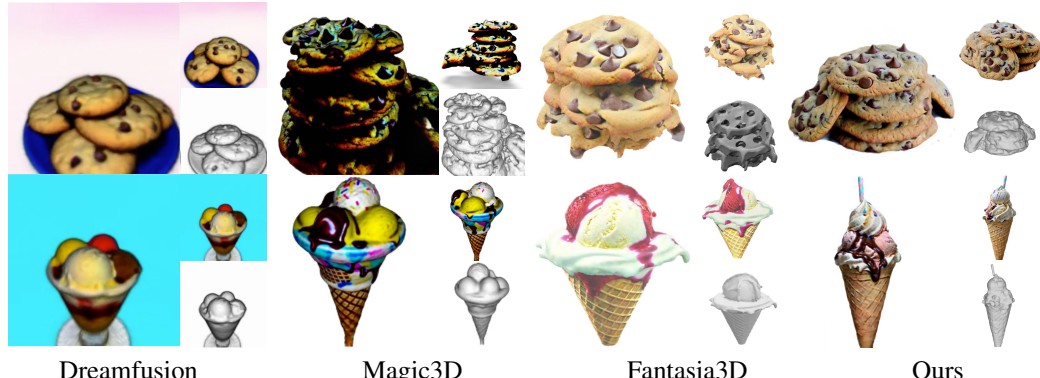

| Dreamfusion | Magic3D | Fantasia3D | Ours |

Figure 4: **Visual comparisons to baseline methods.** We visualize rendered images and extracted meshes, and compare with Dreamfusion (Poole et al., 2022), Magic3D (Lin et al., 2023), and Fantasia3D (Chen et al., 2023b). Prompts: "A plate piled high with chocolate chip cookies" (*top*) and "An ice cream sundae" (*bottom*).

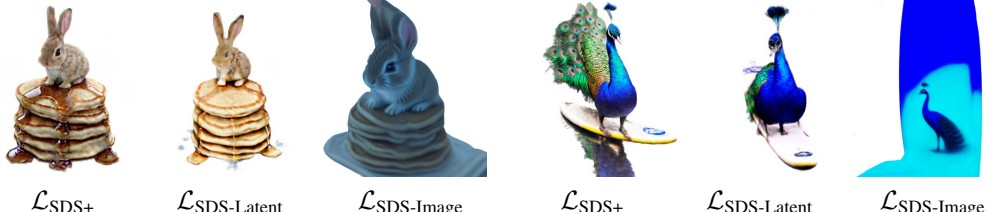

| $\mathcal{L}_{\text{SDS+}}$ | $\mathcal{L}_{\text{SDS-Latent}}$ | $\mathcal{L}_{\text{SDS-Image}}$ | $\mathcal{L}_{\text{SDS+}}$ | $\mathcal{L}_{\text{SDS-Latent}}$ | $\mathcal{L}_{\text{SDS-Image}}$ |

Figure 5: **Ablation study of $\mathcal{L}_{\text{SDS+}}$** with (1) the full SDS loss $\mathcal{L}_{\text{SDS+}}$, (2) the SDS loss in latent space only, denoted as $\mathcal{L}_{\text{SDS-Latent}}$, and (3) the SDS loss in image space only, denoted as $\mathcal{L}_{\text{SDS-Image}}$. Prompts are: (a) "A baby bunny sitting on top of a stack of pancakes" and (b) "A peacock on a surfboard".

Here, $\delta_r$ acts as an indicator function (or binary weight) to filter out background rays. We find this loss remarkably useful for ensuring geometrical consistency and eliminating cloudy geometrical artifacts in our 3D model. In Fig. 6, we also compare the regularization loss $\mathcal{L}_{\text{zvar}}$ to existing regularization strategies (Barron et al., 2022).

Consequently, the total loss function is defined as,

$$\mathcal{L} = \mathcal{L}_{\text{SDS+}} + \lambda_{\text{zvar}}\mathcal{L}_{\text{zvar}}, \tag{9}$$

where $\lambda_{\text{zvar}}$ is the loss weight. We present our training procedure in the appendix, Algorithm 1.

**Kernel smoothing for coarse-to-fine importance sampling.** We observed that while integrating the z-variance loss $\mathcal{L}_{\text{zvar}}$ sharpens the density distribution along the rays, it also intensifies the flickering appearance. We consider the issue arising from the increased challenges of estimating the PDF of volume density along these rays. To address this, we propose a simple yet effective kernel smoothing (KS) technique for coarse-to-fine importance sampling during rendering. Specifically, the KS approach involves a weighted moving average of neighboring PDF values estimated during the coarse stage. The weight is defined by a sliding window kernel. This ensures a broader sampling scope near the high-density regions in the refined stage. Formally, in the coarse stage, for each weight $v_i$ along a ray $r$, the KS approach flattens the weight as $v_i = \frac{\sum_{j=1}^{N} K_j \cdot v_{i+j-\lfloor\frac{N}{2}\rfloor}}{\sum_{j=1}^{N} K_j}$, where $K \in \mathbb{R}^N$ is the kernel. In practice, we set $K = [1, 1, 1]$. In Fig. 3, we visualized in (a) the ground truth and the estimated distribution of volume density along a NeRF ray in the coarse stage. In (b), we display the sampled z-coordinates in the refined stage, either without (on the left) or with (on the right) the KS approach. Their corresponding rendered images are shown in (c) and (e). Fig. 3 shows that the KS approach ensures comprehensive sampling near the peak of the density distribution, achieving multi-view consistent renderings and eliminating flickering issues.

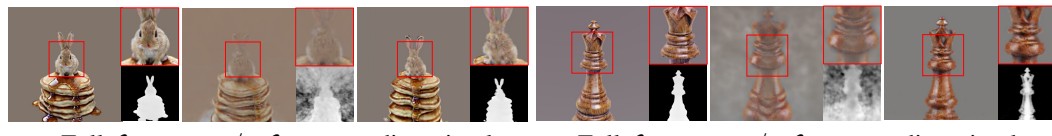

Full $\mathcal{L}$   $w/o\ \mathcal{L}_{\text{zvar}}$   + distortion loss   Full $\mathcal{L}$   $w/o\ \mathcal{L}_{\text{zvar}}$   + distortion loss

Figure 6: **Ablation study of the z-variance loss $\mathcal{L}_{\text{zvar}}$.** We experiment with (1) the full loss $\mathcal{L}$, (2) the loss without the z-variance loss $\mathcal{L}_{\text{zvar}}$, and (3) the loss where the z-variance loss is replaced with an alternative distortion loss (Barron et al., 2022). We show a rendered example on the left, a zoomed-in result at the top right and the corresponding depth image at the bottom right.

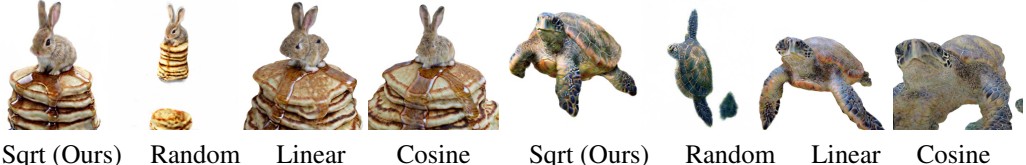

Sqrt (Ours)   Random   Linear   Cosine   Sqrt (Ours)   Random   Linear   Cosine

Figure 7: **Ablation study of timestep annealing.** We experiment various timestep annealing schemes including square root, random sampling, linear and cosine. The results suggest that the square root annealing rate yields supiror performance w.r.t. photo-realism and reasonable geometry.

## 5 EXPERIMENTS

We evaluate our method to generate 3D assets from text prompts in Sec. 5.1. Specifically, we compare our method with popular text-to-3D generation methods, Dreamfusion (Poole et al., 2022), Magic3D (Lin et al., 2023), and Fantasia3D (Chen et al., 2023b). Additional comparisons to the concurrent work ProlificDreamer (Wang et al., 2023b) are presented in Appendix A.4. Moreover, we conduct extensive ablation studies in Sec. 5.2 to verify the effectiveness of each proposed technique. We present experiments with an alternative text-to-image diffusion model in Sec. 5.3. In Sec. 5.4, we extend our method on the image-to-3D reconstruction task, compared with baseline methods (Liu et al., 2023b; Qian et al., 2023; Liu et al., 2023c). Implementation details are in Appendix A.1.

### 5.1 EXPERIMENTAL RESULTS

**Qualitative rendered results** of the 3D assets generated by our approach are depicted in Fig. 1. Our proposed approach generates high-fidelity 3D assets, with photo-realism and multi-view consistency. Additional results are shown in Appendix A.3.

**Qualitative comparisons to baseline methods** are shown in Fig. 4. Specifically, in Fig. 4, we compare our method with Dreamfusion (Poole et al., 2022), Magic3D (Lin et al., 2023), and Fantasia3D (Chen et al., 2023b) for text-to-3D generation. We observe that our rendered images exhibit enhanced photo-realism, improved texture details of the 3D assets, and more natural lighting effects. Additional comparisons are shown in Appendix A.4.

### 5.2 ABLATION STUDY

**Ablation on image-space regularization.** Fig. 5 compares results of three different SDS loss settings: (1) the full SDS loss $\mathcal{L}_{\text{SDS+}}$, (2) the SDS loss in latent space only, denoted as $\mathcal{L}_{\text{SDS-Latent}}$, and (3) the SDS loss in image space only, denoted as $\mathcal{L}_{\text{SDS-Image}}$. The results indicate that incorporating the image-space regularization contributes to a more natural appearance and enhanced texture details, as exemplified by the peacock images. However, relying solely on the image-space loss $\mathcal{L}_{\text{SDS-Image}}$ results in color bias issues, regardless of the guidance scale used.

**Ablation on the z-variance loss $\mathcal{L}_{\text{zvar}}$** is shown in Fig. 6. In this case, we compare the results obtained using (1) the full loss, (2) the loss without the z-variance loss $\mathcal{L}_{\text{zvar}}$, and (3) the loss where z-variance loss $\mathcal{L}_{\text{zvar}}$ is replaced with an alternative distortion loss introduced by Barron et al. (2022), originally for outdoor scene reconstruction. Notably, the absence of the z-variance loss, $\mathcal{L}_{\text{zvar}}$, leads to the generation of assets with cloudy artifacts. While the distortion loss mitigates this cloudiness issue, it occasionally compromises appearance details and hollow geometry. In comparison, our proposed z-variance loss $\mathcal{L}_{\text{zvar}}$ consistently yields photo-realistic results with crisp geometry.

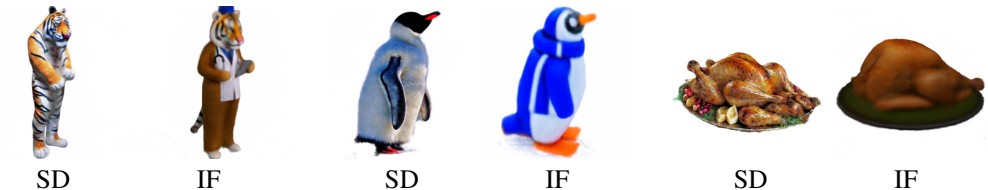

SD        IF        SD        IF        SD        IF

Figure 8: **Experiments with alternative Deep Floyd IF model.** We experiment with the stage-1 model in IF, which employs a T5-XXL (Raffel et al., 2020) text encoder, and provides guidance in $64 \times 64$ resolution. Prompts: (a) "a tiger dressed like a doctor", (b) "a wide angle zoomed out DSLR photo of a skiing penguin wearing a puffy jacket", and (c) "a roast turkey on a platter with only one pair of legs and one pair of wings".

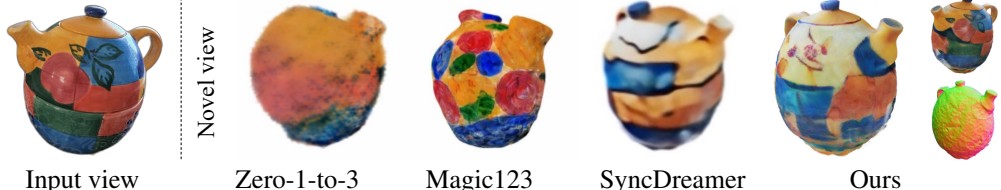

Input view      Zero-1-to-3      Magic123      SyncDreamer      Ours

Figure 9: **Novel view image generation given a single view image**. We compare our method with concurrent works, including Zero-1-to-3 (Liu et al., 2023b), Magic123 (Qian et al., 2023), and SyncDreamer (Liu et al., 2023c).

**Ablation on timestep annealing.** Fig. 7 shows rendering images using different timestep sampling schemes. These include our proposed square root annealing rate, the random sampling adopted in prior works Poole et al. (2022); Lin et al. (2023), and both the linear and cosine annealing rates. We observe that the square root timestep annealing scheme outperforms the other baseline schemes in capturing detailed appearance and geometry.

### 5.3 OPTIMIZATION WITH GUIDANCE FROM AN ADVANCED TEXT ENCODER

In addition to using the SD model (Rombach et al., 2022), we also employ diffusion guidance from the Deep Floyd IF model [1]. The IF model employs an advanced text encoder, T5-XXL (Raffel et al., 2020). In Fig. 8, we show difficulties in generating content for the terms "doctor" and "jacket" and the Janus problems (i.e., multi-face issues) when using the SD model. These challenges can be addressed using the IF model. Here, we only use the stage-1 model in IF, generating images at $64 \times 64$ resolution. A future direction would be to use the full model for high-resolution guidance.

### 5.4 IMAGE-TO-3D RECONSTRUCTION

Our method also enables the reconstruction of 3D assets from a single image. To achieve this, we follow the concurrent work, SyncDreamer (Liu et al., 2023c), which involves hallucinating 16-view images for auxiliary image reconstruction. Then, we incorporate an image (or latent) reconstruction loss for the given view(s) and utilize our score distillation method to optimize the remaining randomly sampled views. Visual comparisons with the baseline methods are in Fig. 9. We observe that our method can produce advancing photo-realistic images from novel views with reasonable details. We also conduct an image-guided hallucination experiment where we initially use the image reconstruction loss, transitioning to exclusively use our proposed loss $\mathcal{L}$ throughout all views. See Appendix A.5 for additional results.

### 6 CONCLUSION

We propose a novel approach for high-quality text-to-3D generation in a single-stage training. Specifically, we distill denoising scores from the pre-trained text-to-image diffusion models in both the image and latent spaces, paired with a novel timestep annealing approach. Moreover, we propose two general improvements for NeRFs, including a z-variance loss and a kernel smooth approach, ensuring 3D representation with consistent appearance and sharp geometry.

---

[1] https://www.deepfloyd.ai/deepfloyd-if

## 7 ACKNOWLEDGEMENTS

This work is partially supported by NSF III 2046795, IIS 1909577, CCF 1934986, NIH 1R01MH116226-01A, NIFA award 2020-67021-32799, the Alfred P. Sloan Foundation, and Google Inc.

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

## A APPENDIX

We present our implementation details in Sec. A.1. Our training algorithm is shown in Sec. A.2. Further 3D asset generation results can be viewed in Sec. A.3, and additional comparisons to text-to-3D baseline methods are available in Sec. A.4. In Sec. A.5, we show additional image-to-3D reconstruction results and experiments with an image-guided hallucination task where we generate 3D assets that are hallucinated from the given input image (rather than reconstruction). Please refer to our video demo in the supplementary material for a comprehensive overview.

### A.1 IMPLEMENTATION DETAILS

**Model setup.** Our approach is implemented based on a publicly available repository [2]. In this implementation, a NeRF is parameterized by a multi-layer perception (MLP), with instant-ngp (Müller et al., 2022) for positional encoding. To enhance photo-realism and enable flexible lighting modeling, instead of using Lambertian shading as employed in (Poole et al., 2022), we encode the ray direction using spherical harmonics and utilize it as an input to NeRF. Additionally, we incorporate a background network that predicts background color solely based on the ray direction. We employ a pre-trained SD model [3] as diffusion prior, as well as a pre-trained dense prediction model [4] to predict disparity maps.

**Training setup.** We use Adam (Kingma & Ba, 2015) with a learning rate of $10^{-2}$ for instant-ngp encoding, and $10^{-3}$ for NeRF weights. In practice, we choose total_iter as $10^4$ iterations. The rendering resolution is $512 \times 512$. We employ DDIM (Song et al., 2021) with empirically chosen parameters $r = 0.25$, and $\eta = 1$ to accelerate training. We choose the hyper-parameters $\lambda_{\text{rgb}} = 0.1, \lambda_d = 0.1$, and $\lambda_{\text{zvar}} = 3$. Similar to prior work (Poole et al., 2022; Lin et al., 2023; Wang et al., 2023a), we use classifier-free guidance (Ho & Salimans, 2022) of 100 for our diffusion model.

### A.2 TRAINING ALGORITHM

We present our training procedure in Algorithm 1. In step 5, either a single-step or multi-step denoising approach can be used to estimate the latent vector $z$. Here, the multi-step denoising refers to the iterative denoising of $\hat{z}_t$, until $t = 0$.

---

**Algorithm 1** Training Procedure

**Input:** A pre-trained SD Rombach et al. (2022) consisting of an encoder $\mathcal{E}$, a decoder $\mathcal{D}$, and a denoising autoencoder $\epsilon_\phi$; a rendering $x = g(\theta)$; a latent vector $z = \mathcal{E}(x)$; a number of total training steps total_iter; range of the diffusion time steps $[t_{\max}, t_{\min}]$; a conditioning $y$; scaling coefficients $\alpha_t$ and $\sigma_t$.

1: **for** iter = [0, total_iter] **do**
2:      $t = t_{\max} - (t_{\max} - t_{\min})\sqrt{\frac{\text{iter}}{\text{total\_iter}}}$
3:      $z_t = \alpha_t z + \sigma_t \epsilon$, where $\epsilon \sim \mathcal{N}(\mathbf{0}, \mathbf{I})$
4:      Estimating noise $\hat{\epsilon} = \epsilon_\phi(z_t; y, t)$
5:      Estimating the latent vector $\hat{z} = \frac{1}{\alpha_t}(z_t - \sigma_t \hat{\epsilon})$ via either single- or multi-step denoising
6:      Estimating the image $\hat{x} = \mathcal{D}(\hat{z})$
7:      Compute the loss gradient $\nabla_\theta \mathcal{L}$ and update $\theta$
8: **end for**
**Return:** $\theta$

---

### A.3 ADDITIONAL RESULTS OF TEXT-TO-3D GENERATION

We provide more generated 3D assets given text prompts in Fig. 10- 12.

---

[2] https://github.com/ashawkey/stable-dreamfusion/tree/main.

[3] We use the pre-trained SD in https://github.com/huggingface/diffusers.

[4] https://github.com/huggingface/transformers.

## A.4 ADDITIONAL COMPARISONS TO THE BASELINE METHODS

We present additional comparisons to the baseline methods in Fig 13- 18, following the rendering settings used in ProlificDreamer (Wang et al., 2023b).

Specifically, in Fig. 13, we present results only using NeRF representation, comparing them to two baseline methods, namely ProlificDreamer (Wang et al., 2023b) and DreamFusion (Poole et al., 2022). In this case, no fine-tuning stage for 3D asset generation is applied in these baseline methods as illustrated in Fig. 13; our method allows the generation of high-fidelity details and natural colors through only a *single-stage* optimization. We observe flickering issues and improper geometries when using only the NeRF representation in ProlificDreamer (Wang et al., 2023b). In contrast, our method consistently provides view and geometry-consistent results without flickering.

In Fig. 17, we present additional visual results, comparing them to the baseline methods, including ProlificDreamer (Wang et al., 2023b), Fantasia3D Chen et al. (2023b), Magic3D (Lin et al., 2023) and DreamFusion (Poole et al., 2022). In this case, the baseline methods (Wang et al., 2023b; Lin et al., 2023) employ the full training pipeline, which includes NeRF representation followed by fine-tuning.

Additional comparisons with Fantasia3D (Chen et al., 2023b) and Magic3D (Lin et al., 2023) are shown in Fig. 14- 16, and comparisons with DreamFusion (Poole et al., 2022) in Fig. 18.

In Fig. 19, we integrate the z-variance loss into ProlificDreamer. We observe that incorporating the z-variance loss results in sharper textures. In Fig.20, we present the results of ProlificDreamer both without and with our proposed method, which includes the z-variance loss, the image-space loss, and the square root time-step annealing schedule. From the results, our method enhances the baseline approach, enabling it to generate superior renderings with detailed textures.

## A.5 ADDITIONAL RESULTS OF IMAGE-TO-3D RECONSTRUCTION

In Fig. 21, we present additional image-to-3D reconstruction results. Additionally, we conduct image-guided 3D hallucination experiments. Specifically, we execute image-to-3D reconstruction at early training iterations, and then optimize the NeRF representation only using our proposed distillation loss, omitting the image reconstruction loss. We show these results in Fig. 22.

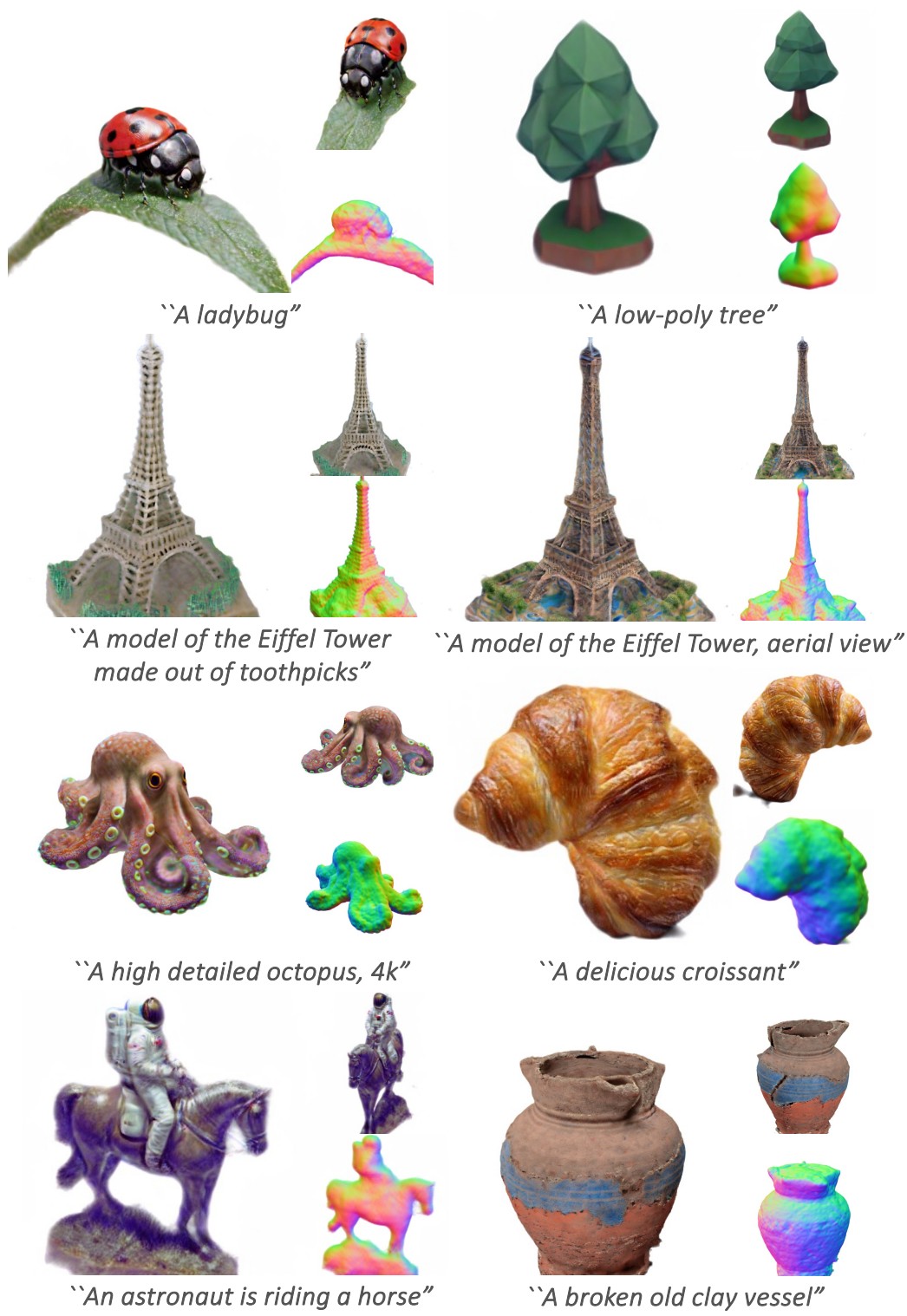

Figure 10: **Additional 3D asset generation results with the corresponding normal map given text prompts (below each object).**

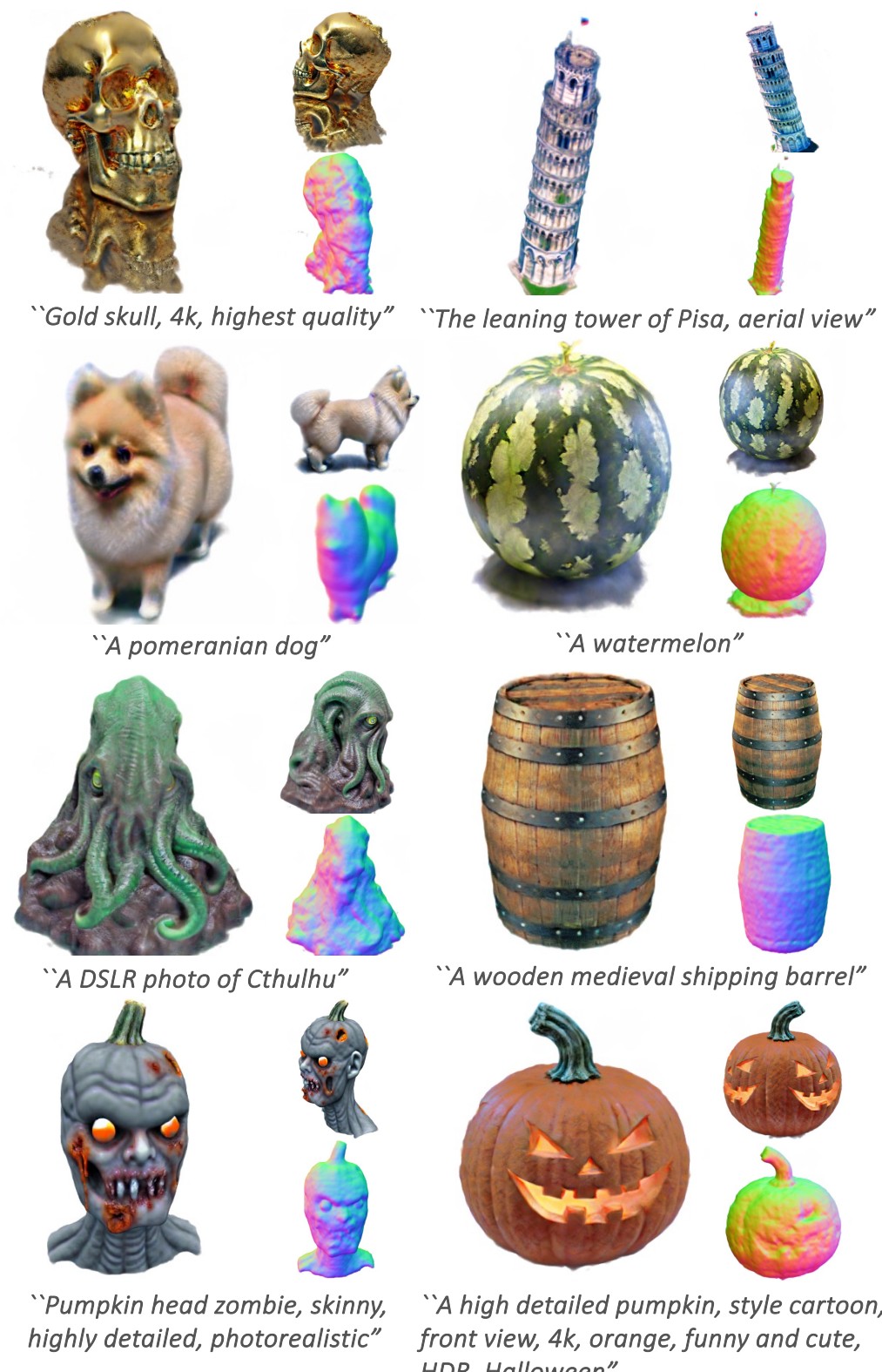

``Gold skull, 4k, highest quality''    ``The leaning tower of Pisa, aerial view''

``A pomeranian dog''    ``A watermelon''

``A DSLR photo of Cthulhu''    ``A wooden medieval shipping barrel''

``Pumpkin head zombie, skinny, highly detailed, photorealistic''    ``A high detailed pumpkin, style cartoon, front view, 4k, orange, funny and cute, HDR, Halloween''

Figure 11: **Additional 3D asset generation results with the corresponding normal map given text prompts (below each object).**

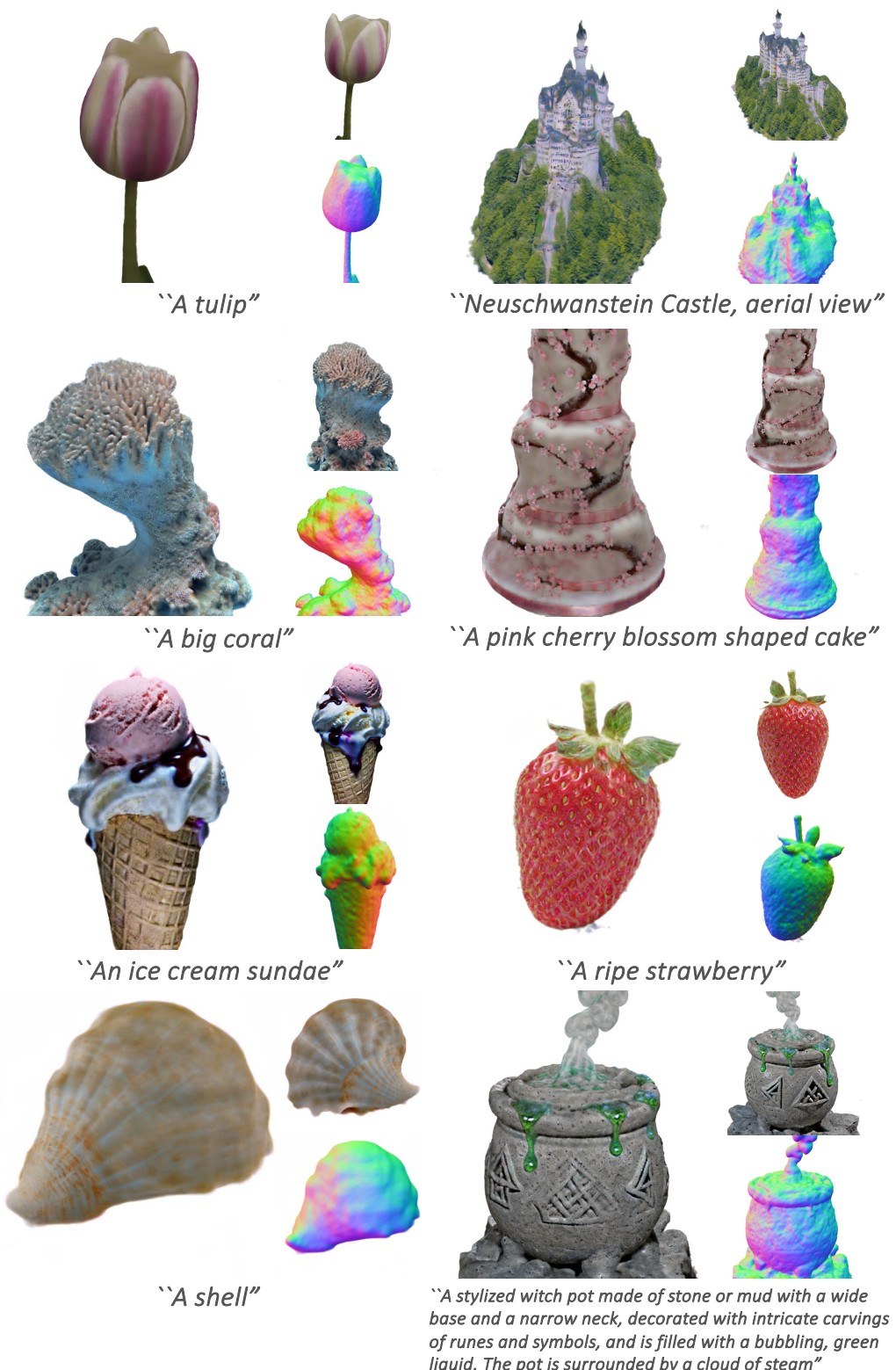

``A tulip"

``Neuschwanstein Castle, aerial view"

``A big coral"

``A pink cherry blossom shaped cake"

``An ice cream sundae"

``A ripe strawberry"

``A shell"

``A stylized witch pot made of stone or mud with a wide base and a narrow neck, decorated with intricate carvings of runes and symbols, and is filled with a bubbling, green liquid. The pot is surrounded by a cloud of steam"

Figure 12: **Additional 3D asset generation results with the corresponding normal map given text prompts (below each object).**

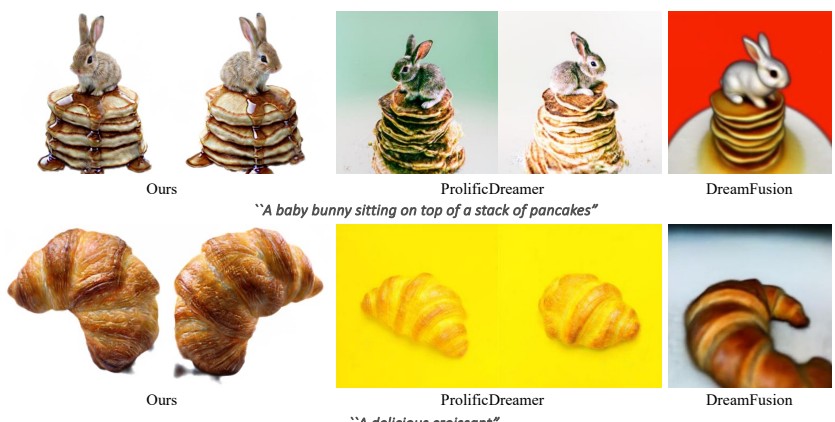

Figure 13: **Additional visual comparisons using NeRF representation only**. We compare visually with the baseline methods, ProlificDreamer (Wang et al., 2023b) and DreamFusion (Poole et al., 2022), specifically after the first training stage. In this case, 3D assets are represented by NeRF, with no additional fine-tuning applied in the baselines.

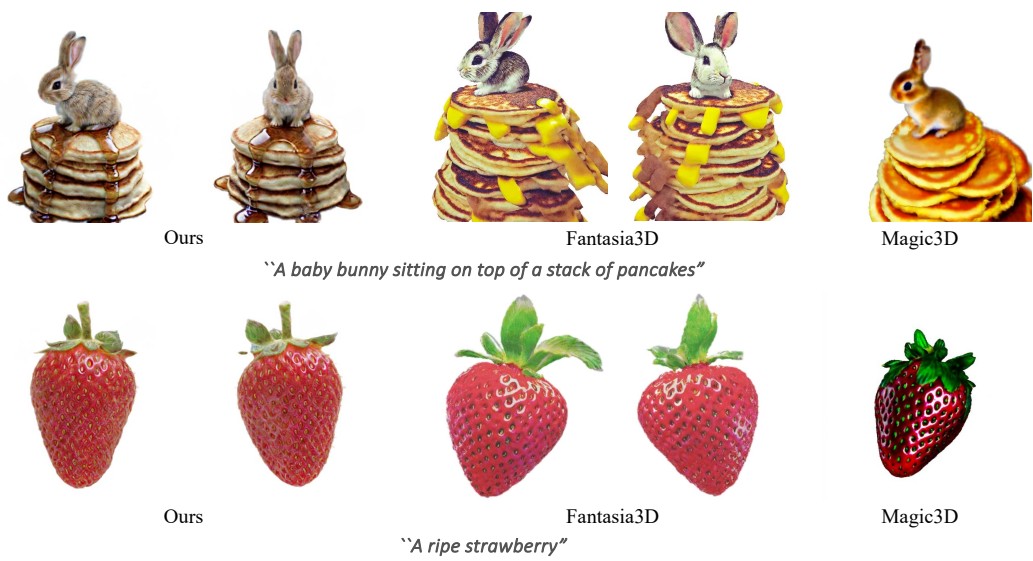

Figure 14: **Additional visual comparisons** with Fantasia3D (Chen et al., 2023b) and Magic3D (Lin et al., 2023).

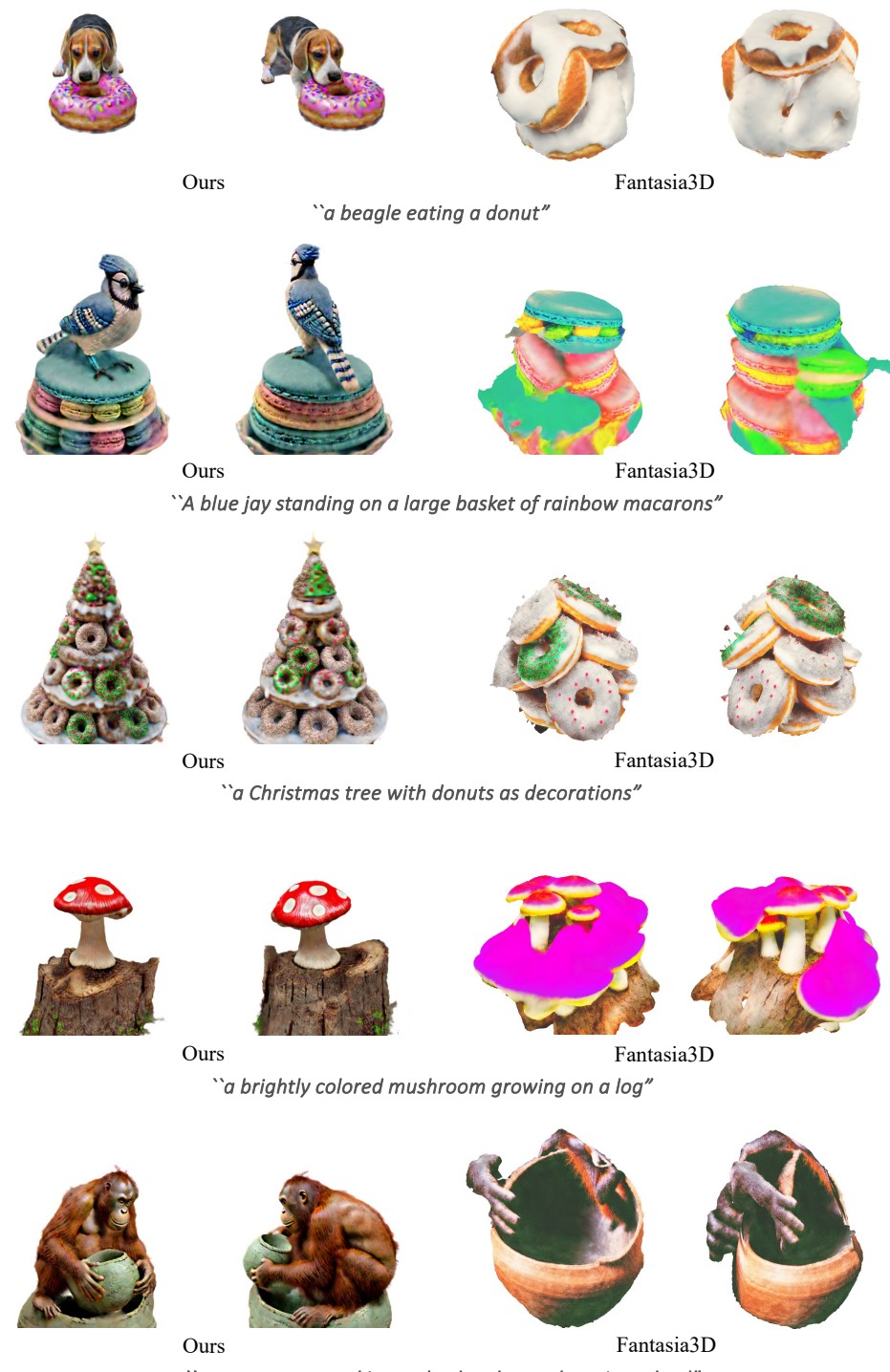

Figure 15: **Additional visual comparisons** with Fantasia3D (Chen et al., 2023b)

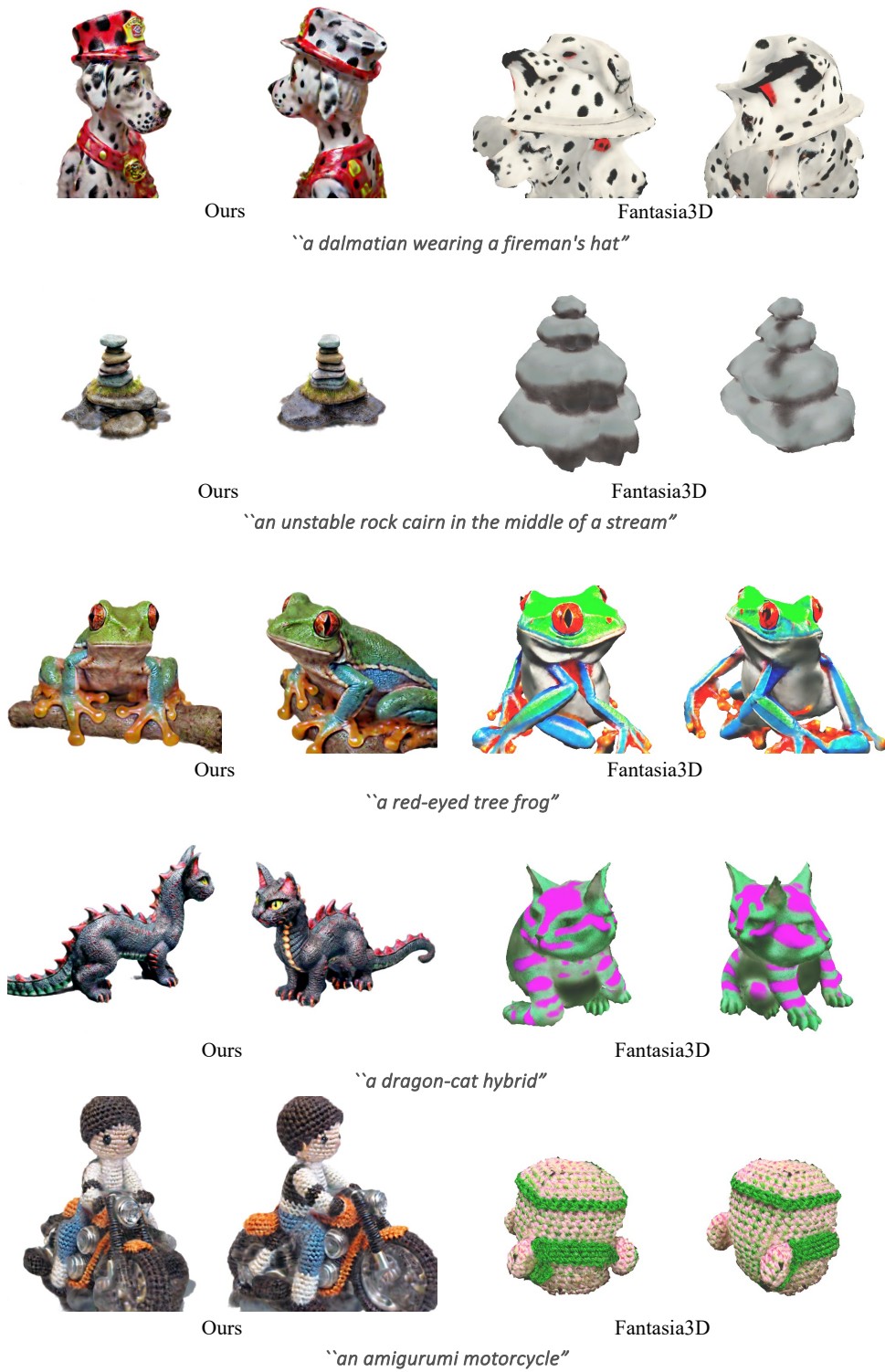

Figure 16: **Additional visual comparisons** with Fantasia3D (Chen et al., 2023b)

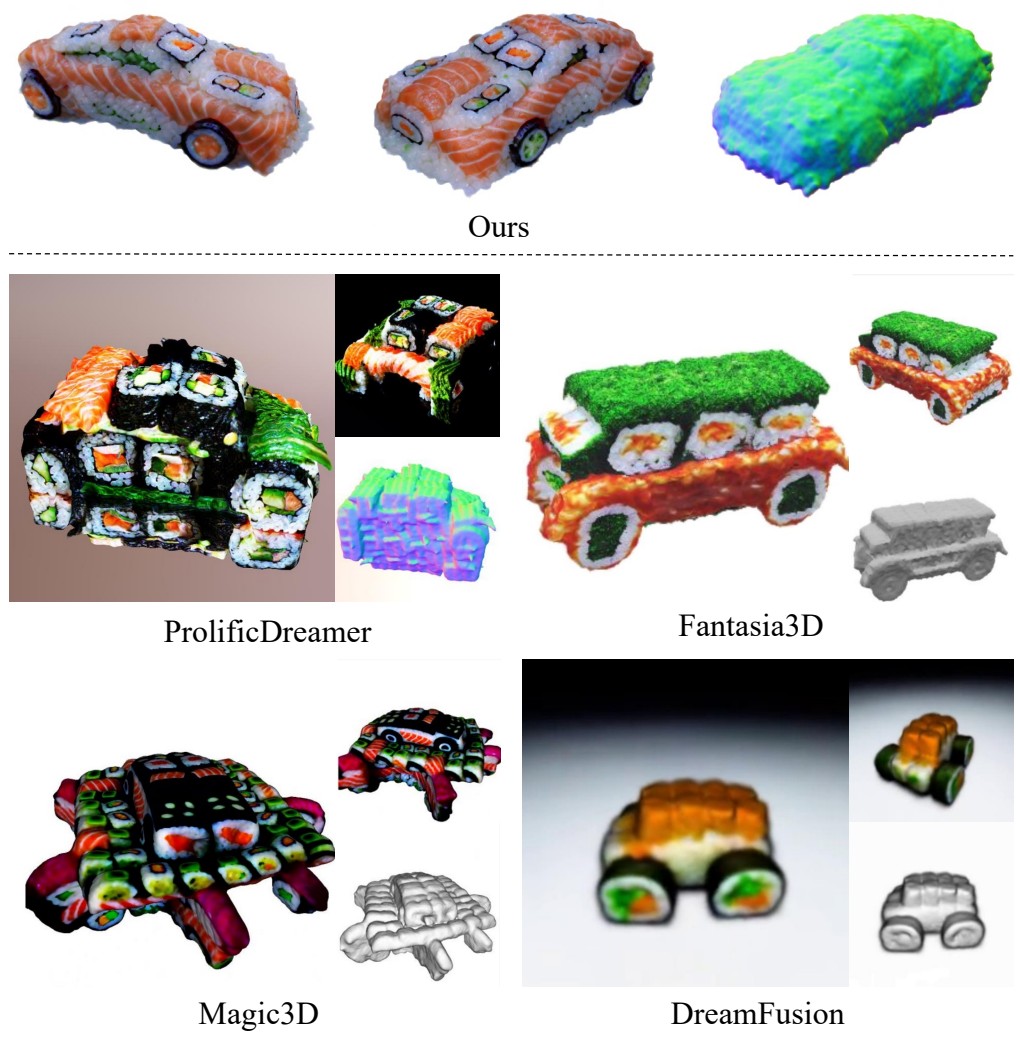

Figure 17: **Additional visual comparisons with the baseline methods**, including Prolific-Dreamer (Wang et al., 2023b), Fantasia3D Chen et al. (2023b), Magic3D (Lin et al., 2023) and DreamFusion (Poole et al., 2022). In this case, the baseline methods (Wang et al., 2023b; Lin et al., 2023) employ the full training pipeline, which includes NeRF representation followed by fine-tuning.

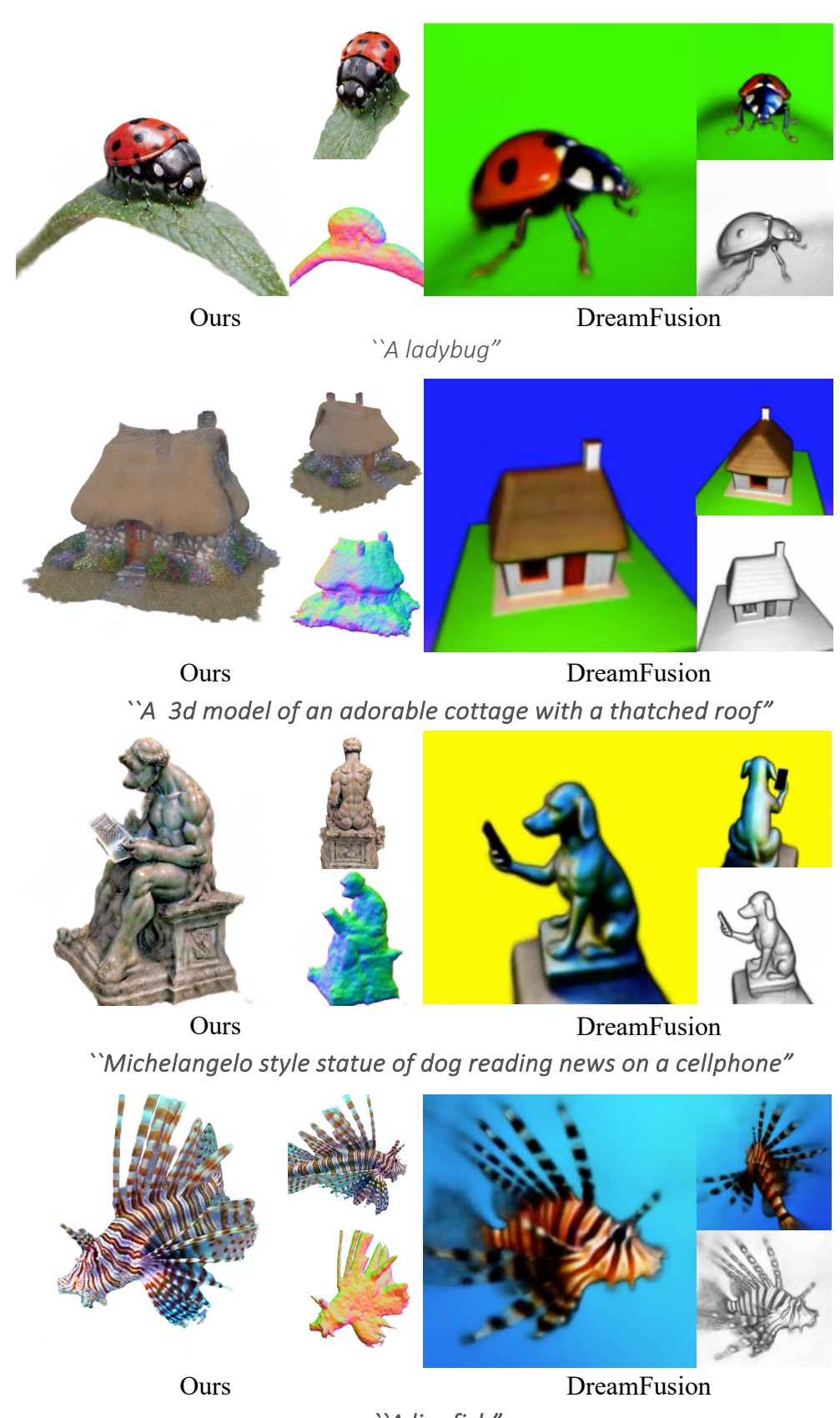

Ours        DreamFusion

``*A ladybug*''

Ours        DreamFusion

``*A 3d model of an adorable cottage with a thatched roof*''

Ours        DreamFusion

``*Michelangelo style statue of dog reading news on a cellphone*''

Ours        DreamFusion

``*A lionfish*''

Figure 18: **Additional visual comparisons** with DreamFusion (Poole et al., 2022).

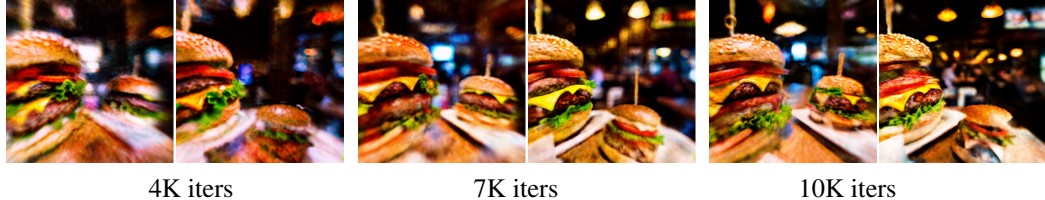

4K iters             7K iters             10K iters

Figure 19: **Visual results of incorporating the z-variance loss to ProlificDreamer (Wang et al., 2023b) throughout the training process.** We show rendered results $w/o$ (left) and $w/$ (right) the z-variance loss after 4K, 7K and 10K training iterations.

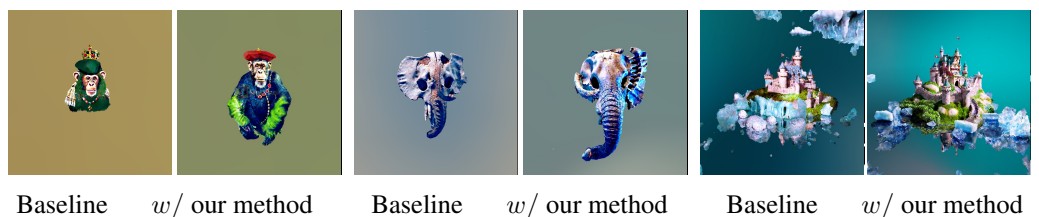

Baseline    $w/$ our method      Baseline    $w/$ our method      Baseline    $w/$ our method

Figure 20: **The baseline results, ProlificDreamer (Wang et al., 2023b), without and with our proposed method.** This includes the z-variance loss, the image-space loss, and the square root time-step annealing schedule.

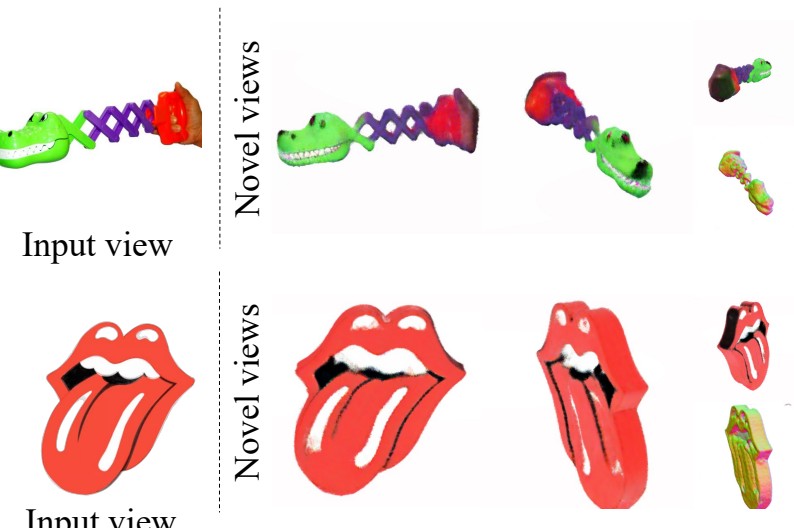

Figure 21: **Additional results of image-to-3D reconstruction.**

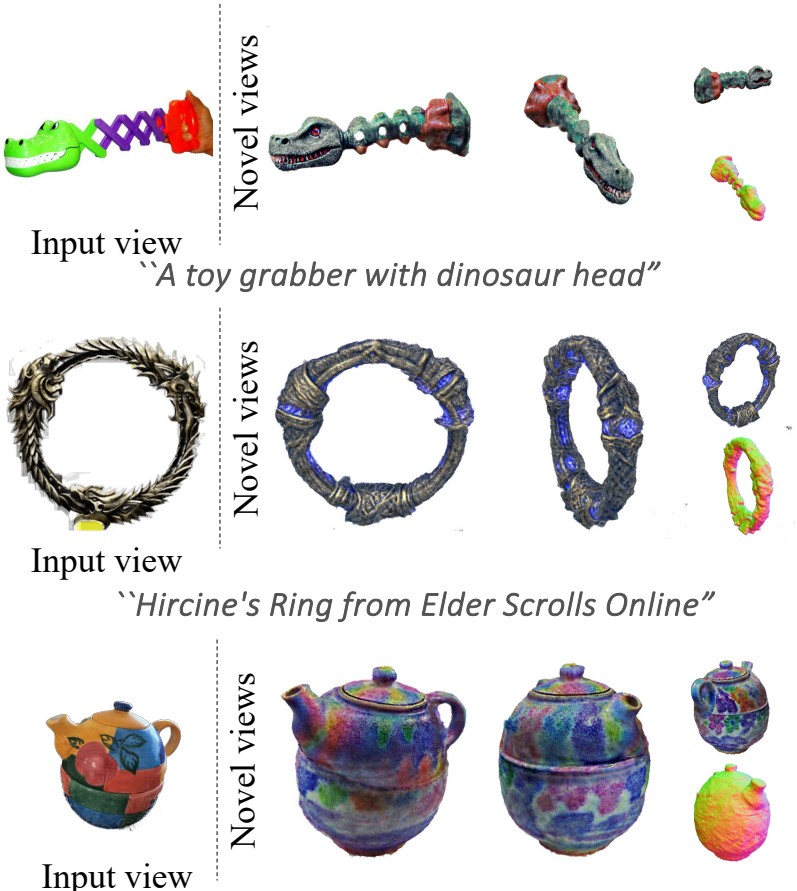

Figure 22: **Visual results of image-guided 3D hallucination.** We **hallucinate** the 3D asset from a single given image using the prompt below the object.

