# OpenReview forum: "HIFA: High-fidelity Text-to-3D Generation with Advanced Diffusion Guidance"
_ICLR.cc/2024/Conference — ICLR 2024 poster_

### Official Review · Reviewer_CJyW · 2023-10-30

**Soundness:** 3 good
**Presentation:** 3 good
**Contribution:** 3 good
**Rating:** 6
**Confidence:** 4

**Summary:**

This work proposes four techniques on the task of 2D diffusion-guided text-to-3D generation, to enhance the generation quality. In particular, the authors 1) propose score distillation in both the latent and image space of the pre-trained text-to-image diffusion models; 2) introduce a timestep annealing strategy to achieve photo-realistic and highly-detailed generation; 3) present a regularization method on the variance of z-coordinates along NeRF rays to encourage crisper surfaces; 4) they also propose a kernel smoothing technique to address flickering issues in the optimized NeRFs. They conduct qualitatively ablation studies and the experimental results demonstrate the effectness of the proposed techniques.

**Strengths:**

* From the presented experimental results (mainly qualitative results), the proposed techniques are effective and improve the performance over prior methods;
* The ablation studies also demonstrate the effectiveness of individual technique;

**Weaknesses:**

* The experimental results are all qualitative results. It is good to have a metric/metrics to compare quantitatively against prior methods; For example, to measure the CLIP similarity between the text prompts and the generated contents; Otherwise, it is difficult to evaluate the performance since we can deliberately select good performing prompts over prior methods for comparisons.

*  The results from Fantasia3D are also very impressive (i.e. in terms of texture quality and geometry) from Figure 4 and Figure 14. Can you provide more results to show that yours is better? I provide following text prompts from DreamFusion: 1) an orangutan making a clay bowl on a throwing wheel; 2) a raccoon astronaut holding his helmet; 3) a blue jay standing on a large basket of rainbow macarons; 4) a corgi taking a selfi; 5) a table with dim sum on it; 6) a lion reading the newspaper; 7) a tiger dressed as a doctor; 8) a chimpanzee dressed like Henry VIII king of England; 9) an all-utility vehicle driving across a stream; 10) a squirrel gesturing in front of an easel showing colorful pie charts. Can you do the comparisons with those prompts?

* For the kernel smoothing, you only choose [1, 1, 1] as the sliding window kernel, have you tried other choices?

**Questions:**

* How is the kernel smoothing conducted for coarse-to-fine importance sampling? Could the authors provide more details? In fact, I do not understand "kernel smoothing". Use an equation to explain it would be very helpful. Figure 3 seems only presents the results with/without kernel smoothing.

---

> ### Author Response · Authors · 2023-11-21
> **Rebuttal to R3**
>
> **Q: “Can you provide more results to Fantasia3D with my given prompts from DreamFusion.”**
>
> For a comprehensive evaluation, both qualitative and quantitative, we obtained 30 text prompts from DreamFusion and conducted experiments using our method and Fantasia3D. We followed the training instructions from the official repository of Fantasia3D.
>
> **Qualitative comparison**: Additional visual comparisons to Fantasia3D were included in Fig.14-15 (10 samples) of the Appendix. We also added a comparison of the rendered videos (``fantasia3d.mp4”, 30 samples) in the supplementary material. Notably, we observed a low success rate in the geometry generation stage using Fantasia3D, even with a higher number of training iterations. We attribute this to the increased difficulty of learning DMTet with the SDS loss, especially when compared to other implicit representations such as NeRFs.
>
> **Quantitative comparison**: We added a user study and computed the CLIP similarity for evaluation:
>
> **(1) User study**: We conducted a user study for our method and Fantasia3D. In the survey, we present rendered 2D images of the 30 generated 3D objects. Users are asked to choose the result that, in their opinion, exhibits overall better quality. Below, we report the macro-average rate of preference for each method across the 30 objects. The results indicate that our method achieves a higher preference than Fantasia3D in terms of overall quality.
>
> | Method 1 | Preference 2 |
> |----------|----------|
> | Fantasia3D | 9.7% |
> | Ours | 90.3% |
>
>
>
> **(2) CLIP similarity**: We also compare our method to Fantasia3D using CLIP-Similarity (↑). In this evaluation, we render 100 images for each generated object and compute the averaged CLIP similarity for the text-image pairs. We use the model "openai/clip-vit-base-patch16” for this evaluation. The scores show that our method achieves slightly better clip similarity than Fantasia3D.
>
> | Method 1 | CLIP-Similarity(↑) |
> |----------|----------|
> | Fantasia3D | 0.302 |
> | Ours | 0.344 |
>
> We have incorporated these evaluations into the manuscript, highlighted in the red color.
>
> **Q: “It is good to have a metric/metrics to compare quantitatively against prior methods; For example, to measure the CLIP similarity”**
>
> Please refer to the answer to the above question.
>
>
> **Q:“For the kernel smoothing, you only choose [1, 1, 1] as the sliding window kernel, have you tried other choices?”**
>
> In addition to using the proposed kernel [1, 1, 1], we also experimented with alternative weighted kernels. For example, we set the kernel K = scale_k * [1, 6, 1], where scale_k represents a scale parameter, and [1, 6, 1] can be interpreted as the importance weights assigned to the current signal and its neighbors within the kernel window. We also varied the weights [1, 6, 1] to explore different proportions. Our experiments showed no significant differences. Notably, as the PDF estimated from the coarse stage has broad coverage in non-zero density regions, and the number of sampling points is fair (e.g., 36 points per ray, compared to 64-128 points used in prior works) during the refined stage, the rendered output does not contain flickering.
>
>
> **Q: “How is the kernel smoothing conducted for coarse-to-fine importance sampling? Could the authors provide more details?”**
>
> **Motivation:**
>
> Recall that NeRFs usually use a hierarchical sampling procedure with two distinct MLPs, one “coarse” and one “fine.” The coarse MLP captures the broad information of the scene, and the fine MLP captures detailed features. This is necessary for NeRF because the MLPs were only able to learn a model of the scene for one single scale (either the "coarse” or the "fine” scale) [1]. However, using two MLPs can be expensive.
>
> *Thus, how to use a single MLP that can learn multiscale representations of the scene?* To address this, we propose a kernel smoothing (KS) approach. The KS approach involves flattening the estimated density in the coarse stage, enabling it to capture a broad signal range of the scene.
>
> **Details:**
>
> Specifically, the KS approach is a weighted moving average of neighboring signals. The weight is defined by a kernel. In practice, for each estimated weight point v_i in the coarse stage, we choose the kernel window as N and compute a weighted average for all signals within the kernel window:
>
> >v_i = \frac{\sum_{j=1}^{N} K_j \cdot v_{i+j- \lfloor \frac{N}{2} \rfloor}}{\sum_{j=1}^{N} K_j}
>
> In practice, we set K = [1,1,1].
>
> **Paper modification and code release**:
>
> We have included a detailed explanation of the KS approach in Sec.4.2 of the manuscript. Additionally, we will provide code implementation with the final version.
>
> [1] Jon Barron et al., Mip-NeRF: A Multiscale Representation for Anti-Aliasing Neural Radiance Fields, ICCV 2021.

---

### Official Review · Reviewer_WWao · 2023-10-30

**Soundness:** 4 excellent
**Presentation:** 3 good
**Contribution:** 3 good
**Rating:** 8
**Confidence:** 3

**Summary:**

- The paper proposes holistic sampling and smoothing approaches for high-quality text-to-3D generation in a single-stage optimization.
- The method introduces a timestep annealing approach and regularization for the variance of z-coordinates along NeRF rays.
- The paper also addresses texture flickering issues in NeRFs with a kernel smoothing technique.
- Experiments show the method's superiority over previous approaches.

**Strengths:**

- The single-stage optimization is useful to the generation of highly detailed and view-consistent 3D assets and the proposed solution to it is impressive.
- Compared to baseline methods like Dreamfusion, Magic3D, and Fantasia3D, the rendered images from this approach exhibit enhanced photo-realism, improved texture details of the 3D assets, and more natural lighting effects.

**Weaknesses:**

- The paper could benefit from a more explicit explanation in the introduction regarding why previous works were unable to achieve single-stage optimization.
- The contributions presented in the paper seem fragmented, lacking a cohesive thread or central theme. It would enhance the paper's clarity and impact if the authors could refine the structure.

**Questions:**

I believe the technical aspects are articulated clearly and technically sound. Thus, I have no further questions.

---

> ### Author Response · Authors · 2023-11-21
> **Rebuttal to R2**
>
> **Q: “The paper could benefit from a more explicit explanation in the introduction regarding why previous works were unable to achieve single-stage optimization.”**
>
> Thanks for the useful feedback. We modified the Sec.1 of the manuscript according to the reviewer’s suggestion. Specifically, we added
>
> >``Moreover, generating a detailed 3D asset through single-stage optimization is challenging. Specifically, explicit 3D representations, such as meshes, struggle to capture intricate topology, such as objects with holes. Implicit 3D representations, such as NeRF, may lead to cloudy geometry and flickering textures.“
>
>
> **Q: “The contributions presented in the paper seem fragmented, lacking a cohesive thread or central theme. It would enhance the paper's clarity and impact if the authors could refine the structure.”**
>
> We thank the reviewer for the useful feedback. We believe that the innovative idea of the paper is well-organized when viewed at a systemic level, aiming to enhance the entire text-to-3D generation system by refining its two essential components: **(1) representation** and **(2) supervision**. These improvements result from a synergistic fusion of various methods. As a result, our contributions stand as comprehensive improvements across every component of the system, incorporating nuanced ``look-like” discrete details.
>
> Specifically, within the framework of text-to-3D generation, we  enhance generation quality for representation and supervision:
> - For representation, we introduce effective regularizations and an advanced importance sampling approach in NeRFs.
> - For supervision, we propose a novel training schedule and an advanced loss function for score distillation.
>
> We reorganized our manuscript to enhance its structure and emphasize the contributions for the entire text-to-3D framework.

---

### Official Review · Reviewer_iKQ3 · 2023-10-31

**Soundness:** 3 good
**Presentation:** 3 good
**Contribution:** 3 good
**Rating:** 6
**Confidence:** 4

**Summary:**

This paper proposed an improved version of Score Distillation Sampling by introducing several strategies. The authors proposed to perform denoising in both image and latent space for better performance. A novel timestep annealing strategy is provided to reduce the sampling space. Besides, the authors also provide a z-coordinates regularization term to achieve high-quality rendering in a single-stage optimization. The paper is well-organized and easy to follow. The proposed strategies are effective to improve the performance.

**Strengths:**

1. The strategy by denoising in both image and latent space is useful to improve the details.
2. The proposed z-variance loss alleviates cloudy artifacts and shows better performance than distortion loss.
3. The proposed method achieves high-quality text-to-3D generation.

**Weaknesses:**

Basically the paper is good and I have several concerns:
1. The contributions of the paper are discrete, which comprises several small contribution points.
2. Maybe a user-study should be conducted to quantitatively evaluate the method.
3. Basically the proposed strategies can be generalized to other baseline methods, for example, ProlificDreamer [1]. I’m curious to see the performance with other baseline methods.

[1] Wang, Z., Lu, C., Wang, Y., Bao, F., Li, C., Su, H., & Zhu, J. (2023). ProlificDreamer: High-Fidelity and Diverse Text-to-3D Generation with Variational Score Distillation. arXiv preprint arXiv:2305.16213.

**Questions:**

Please refer to weaknesses.

---

> ### Author Response · Authors · 2023-11-21
> **Rebuttal to R1**
>
> **Q: “The contributions of the paper are discrete, which comprises several small contribution points.”**
>
> We thank the reviewer for the useful feedback. We believe that the innovative idea of the paper is well-organized when viewed at a systemic level,: aiming to enhance the entire text-to-3D generation system by refining its two essential components: **(1) 3D representation** and **(2) optimization**. These improvements result from a synergistic fusion of various methods. As a result, our contributions stand as comprehensive improvements across every component of the system, incorporating nuanced ``look-like” discrete details under a solid base insight.
>
> Specifically, within the framework of text-to-3D generation, we  enhance generation quality for both representation and optimization:
> - For 3D representation, we introduce effective regularizations and an advanced importance sampling approach in NeRFs.
> - For optimization, we propose a novel training schedule and an advanced loss function for score distillation.
>
> We reorganized our manuscript to enhance its structure and emphasize the contributions for the entire text-to-3D framework.
>
> **Q: “Maybe a user-study should be conducted to quantitatively evaluate the method.”**
>
> For a more comprehensive evaluation, both qualitative and quantitative, we obtained 30 text prompts from DreamFusion and conducted experiments using our method and Fantasia3D. We followed the training instructions from the official repository of Fantasia3D.
>
> **Qualitative comparison**: Additional visual comparisons to Fantasia3D were included in Fig.14-15 (10 samples) of the Appendix. We also added a comparison of the rendered videos (``fantasia3d.mp4”, 30 samples) in the supplementary material. Notably, we observed a low success rate in the geometry generation stage using Fantasia3D, even with a higher number of training iterations. We attribute this to the increased difficulty of learning DMTet with the SDS loss, especially when compared to other implicit representations such as NeRFs.
>
> **Quantitative comparison**: We added a user study and computed the CLIP similarity for evaluation:
>
> **(1) User study**: We conduct a user study for our method and Fantasia3D. In the survey, we present rendered 2D images of the 30 generated 3D objects. Users are asked to choose the result that, in their opinion, exhibits overall better quality. Below, we report the macro-average rate of preference for each method across the 30 objects. The results indicate that our method achieves a higher preference than Fantasia3D in terms of overall quality.
>
> | Method | Preference |
> |----------|----------|
> | Fantasia3D | 9.7% |
> | Ours | 90.3% |
>
>
> **(2) CLIP similarity**: We also compare our method to Fantasia3D using CLIP-Similarity(↑). In this evaluation, we render 100 images for each generated object and compute the averaged CLIP similarity for the text-image pairs. We use the model "openai/clip-vit-base-patch16” for this evaluation. The scores show that our method achieves slightly better clip similarity than Fantasia3D.
>
> | Method | CLIP-Similarity (↑)|
> |----------|----------|
> | Fantasia3D | 0.302 |
> | Ours | 0.344 |
>
>
> We have incorporated these evaluations into the manuscript, highlighted in the red color.
>
>
> **Q: “The proposed strategies can be generalized to other baseline methods, for example, ProlificDreamer.”**
>
> We have integrated the z-variance loss into ProlificDreamer, and the results are presented in Fig.19 in the appendix. Our observations indicate that adding the z-variance loss leads to sharper textures, highlighting the effectiveness and generalizability of the approach.
>
> Additionally, our proposed latent- and image-space SDS loss has been incorporated in various image-to-3D [1] works, particularly in cases where image-space supervision is necessary. Our SDS loss has proven effective in addressing over-saturation issues in these contexts.
>
> [1] Huang et al., HumanNorm: Learning Normal Diffusion Model for High-quality and Realistic 3D Human Generation, 2023.

---

> > ### Author Response · Authors · 2023-11-23
> > **Additional experiments for incorporating our method to ProlificDreamer**
> >
> > **Q: “The proposed strategies can be generalized to other baseline methods, for example, ProlificDreamer.”**
> >
> > We also incorporate our proposed method into ProlificDreamer including the z-variance loss, the image-space loss and the square root timestep annealing schedule. We used the unofficial implementation of ProlificDreamer (https://github.com/threestudio-project/threestudio) due to the absence of official code release. The results are shown in Fig.20 in the appendix.pdf. We observe that that the incorporation of our method into ProlificDreamer significantly improves generation quality, enhancing both detailed textures and sharper shapes.

---

### Author Response · Authors · 2023-11-21
**To all reviewers**

We thank all reviewers for their thoughtful feedback. **We modified the manuscript based on the suggestions.** We also replied to all the reviewers' questions in detail and included corresponding discussions and experiments in the revised submission, which we hope reviewers take into consideration in their final assessment. **We have highlighted the updated content in red color.** We thank the reviewers for their time. In summary, reviewers have highlighted the following aspects of our submission:
- iKQ3: “The proposed method achieves high-quality text-to-3D generation”. “The proposed strategies are effective to improve the performance”.
- WWao: “The single-stage optimization is useful to the generation of highly detailed and view-consistent 3D assets and the proposed solution to it is impressive”. “Compared to baseline methods,  the rendered images from this approach **exhibit enhanced photo-realism, improved texture details** of the 3D assets, and **more natural lighting effects**”.
- CJyW: “The proposed techniques are effective and improve the performance over prior methods”

---

### Meta-Review · Area_Chair_NbGQ · 2023-12-10

**Metareview:**

The paper introduces an improved Score Distillation Sampling method, which incorproates novel strategies to enhance denoising processes in both image and latent spaces. This paper has been reviewed by three experts in the field.

All the reviewers have recognized a key strength of the paper: the implementation of single-stage optimization has been particularly effective in generating 3D assets that are not only highly detailed but also consistent across different views. Such a contribution is a notable advancement in the field and underscores the paper's technical merit.

Reflecting on the positive and insightful feedback from the reviewers, I am pleased to recommend this paper for acceptance at ICLR 2024. It is important to note that the reviewers have also presented some minor concerns. These concerns should be carefully addressed in the final camera-ready version of the manuscript. The modifications are essential to ensure the highest quality and clarity of the final publication. Congratulations again to the authors on the acceptance of their paper.

**Justification For Why Not Higher Score:**

The quantitative comparisons are not comprehensive enough.

**Justification For Why Not Lower Score:**

The strategy by denoising in both image and latent space is useful to improve the details.

---

### Decision · Program_Chairs · 2024-01-16

Accept (poster)